# SIV Infection and the HIV Proteins Tat and Nef Induce Senescence in Adipose Tissue and Human Adipose Stem Cells, Resulting in Adipocyte Dysfunction

**DOI:** 10.3390/cells9040854

**Published:** 2020-04-01

**Authors:** Jennifer Gorwood, Tina Ejlalmanesh, Christine Bourgeois, Matthieu Mantecon, Cindy Rose, Michael Atlan, Delphine Desjardins, Roger Le Grand, Bruno Fève, Olivier Lambotte, Jacqueline Capeau, Véronique Béréziat, Claire Lagathu

**Affiliations:** 1Lipodystrophies, Metabolic and Hormonal Adaptation, and Aging, UMR_S 938, Centre de Recherche Saint-Antoine-Institut Hospitalo-Universitaire de Cardiométabolisme et Nutrition (ICAN), INSERM, Sorbonne Université, F-75012 Paris, France; jennifer.gorwood@inserm.fr (J.G.); tina.ejlalmanesh@inserm.fr (T.E.); matthieu.mantecon@inserm.fr (M.M.); cindy.rose@inserm.fr (C.R.); drmichaelatlan@gmail.com (M.A.); bruno.feve@inserm.fr (B.F.); jacqueline.capeau@inserm.fr (J.C.); 2Immunology of Viral infections and Autoimmune Diseases, IDMIT Department, IBFJ, U1184, INSERM-CEA-Université Paris Sud 11, F-92260 Fontenay-Aux-Roses and F-94270 Le Kremlin-Bicêtre, France; christine.bourgeois@universite-paris-saclay.fr (C.B.); olivier.lambotte@aphp.fr (O.L.); 3Plastic Surgery Department, Tenon Hospital, AP-HP, F-75020 Paris, France; 4IDMIT Department, Center for Immunology of Viral Infections and Autoimmune Diseases, Inserm, CEA, Université Paris Saclay, F-92260 Fontenay-aux-Roses, France; delphine.desjardins@cea.fr (D.D.); roger.le-grand@cea.fr (R.L.G.); 5Diabétologie et Reproduction, PRISIS, Service d’Endocrinologie, Hôpital Saint-Antoine, AP-HP, F-75012 Paris, France; 6Service de Médecine Interne et Immunologie Clinique, Groupe Hospitalier Universitaire Paris Sud, Hôpital Bicêtre, AP-HP, F-94270 Le Kremlin-Bicêtre, France

**Keywords:** adipose stem cells, HIV, senescence, oxidative stress, adipogenesis

## Abstract

Background: Aging is characterized by adipose tissue senescence, inflammation, and fibrosis, with trunk fat accumulation. Aging HIV-infected patients have a higher risk of trunk fat accumulation than uninfected individuals—suggesting that viral infection has a role in adipose tissue aging. We previously demonstrated that HIV/SIV infection and the Tat and Nef viral proteins were responsible for adipose tissue fibrosis and impaired adipogenesis. We hypothesized that SIV/HIV infection and viral proteins could induce adipose tissue senescence and thus lead to adipocyte dysfunctions. Methods: Features of tissue senescence were evaluated in subcutaneous and visceral adipose tissues of SIV-infected macaques and in human adipose stem cells (ASCs) exposed to Tat or Nef for up to 30 days. Results: p16 expression and p53 activation were higher in adipose tissue of SIV-infected macaques than in control macaques, indicating adipose tissue senescence. Tat and Nef induced higher senescence in ASCs, characterized by higher levels of senescence-associated beta-galactosidase activity, p16 expression, and p53 activation vs. control cells. Treatment with Tat and Nef also induced oxidative stress and mitochondrial dysfunction. Prevention of oxidative stress (using N-acetyl-cysteine) reduced senescence in ASCs. Adipocytes having differentiated from Nef-treated ASCs displayed alterations in adipogenesis with lower levels of triglyceride accumulation and adipocyte marker expression and secretion, and insulin resistance. Conclusion: HIV/SIV promotes adipose tissue senescence, which in turn may alter adipocyte function and contribute to insulin resistance.

## 1. Introduction

Aging is a major issue in the general population; it is associated with adipose tissue senescence, inflammation, and fibrosis, leading to trunk fat redistribution [1,2,3]. It has been suggested that aging contributes to the overall metabolic and functional decline of adipose tissue.

Given that highly effective antiretroviral therapy (ART) can control but not cure HIV infections, HIV-infected patients are aging with the virus. Moreover, the prevalence of comorbidities typically associated with aging (e.g., cardiometabolic diseases) is higher in people living with HIV that in the general population. In particular, the risk of trunk fat accumulation (associated with pro-atherogenic metabolic factors) is greater in aging HIV-infected people than in uninfected controls, and this risk increases with age [2,4,5]. The pathophysiological mechanisms underlying adipose tissue redistribution in aging HIV-infected people are poorly understood but are almost certainly multifactorial. Earlier research has suggested that both the HIV infection itself and some ARTs have roles in adipose tissue inflammation and extracellular matrix remodeling [6,7,8,9]. Adipose tissue aging has not been investigated previously in the context of SIV/HIV infection. Therefore, we sought to determine whether or not SIV/HIV-infected adipose tissue displayed features of accentuated aging that could contribute to cardiometabolic disorders.

Cellular aging (also referred to as cellular senescence) is defined as an irreversible growth arrest [10] and an increase in levels of cell cycle arrest proteins (including p16, p21, and phosphorylated-p53) [11]. Senescent cells are characterized by elevated levels of senescence-associated (SA)-β-galactosidase activity, and foci of DNA damage [12]. Furthermore, these cells display a specific, pro-inflammatory, senescence-associated secretory phenotype (SASP) [13]. Senescence can result not only from excessive oxidative stress, mitochondrial dysfunction, inflammation, and DNA damage but also from environmental stimuli, such as viral infections [11,14].

In the context of HIV infection, the results of previous in vitro studies have shown that some ARTs (such as protease inhibitors and nucleoside reverse transcriptase inhibitors) can induce senescence in human and murine mesenchymal stem cells (MSCs) [15,16,17]. Elevated levels of p16 and prelamin A (indicating senescence) have been observed in the adipose tissue of ART-treated HIV-infected people, and may be related to the effect of protease inhibitors (PIs) [3]. However, few studies have looked at the impact of SIV/HIV on cellular senescence. We have reported that the viral proteins trans-activator of transcription (Tat) and negative-regulating factor (Nef) induce cellular senescence in bone marrow MSCs via the NF-κB [18] and autophagy [19] pathways, respectively.

Subcutaneous adipose tissue (SCAT) and visceral adipose tissue (VAT) are known to be viral reservoirs that notably contain HIV-infected CD4^+^ T cells and macrophages [20,21]. Thus, within adipose tissue, viral proteins released by infected cells might act on nearby adipocytes and stroma-vascular cells, including adipose precursor cells and adipose stem cells (ASCs). In adipose tissue, ASCs are involved in renewal of the pool of adipocytes through their recruitment for adipogenesis. It has been suggested that the senescence-induced exhaustion of ASCs can disrupt adipose tissue homeostasis [1,22]. Taken as a whole, these findings suggest that HIV infection is involved in ASC senescence, which in turn might contribute to the adipose tissue redistribution and alterations observed in HIV-infected people.

Therefore, the objective of the present study of two complementary in vivo and in vitro models was to evaluate the impact of HIV/SIV on adipose tissue aging, inflammation, and oxidative stress. First, we revealed the impact of the virus per se on adipose tissue aging in vivo in a novel nonhuman primate model of chronic SIV infection. Then, we showed that the HIV proteins Tat and Nef released by infected cells can induce senescence in human ASCs in vitro—probably through the induction of oxidative stress. Nef-induced senescence impacted the adipogenic capacity of ASCs and resulted in insulin resistance—suggesting that adipose tissue senescence contributes to the fat alterations and cardiometabolic complications observed in HIV-infected people.

## 2. Materials and Methods

### 2.1. Macaque Infection and Sample Collection

Adult cynomolgus macaques (*Macaca fascicularis*) were imported from Mauritius, handled, and housed in the animal facilities at the *Commissariat à l’Energie Atomique et aux Energies Alternatives* (CEA, Fontenay-aux-Roses, France; CEA Permit Number A 92-032-02). The CEA animal facilities comply with the Standards for Human Care and Use of Laboratory of the Office for Laboratory Animal Welfare (OLAW, USA, assurance number #A5826-01) and with the European Directive (2010/63, recommendation No. 9). The study was authorized by the local animal care and use committee (*Comité d’éthique en expérimentation animale* no. 44: Reference: 2015102713323361.02, APAFIS#2453) and the French Ministry of Research (*Ministère de l’Enseignement Supérieur et de la Recherche*). Macaques were intravenously infected (or not, for controls) with SIVmac251, as described previously [9,20]. The mean ± SEM duration of SIV infection was 1.3 ± 0.3 years (Appendix A). At sacrifice, animals were sedated with ketamine chlorhydrate, and then euthanized by intravenous injection of sodium pentobarbital (Vetoquinol, Paris, France). The dates of euthanasia and necropsy of control and SIV-infected animals were programmed in the context of specific studies using these animals. Samples of abdominal subcutaneous adipose tissue (SCAT) and visceral adipose tissue (VAT) were collected at necropsy. Non-adipose-associated tissues (including lymph nodes and blood vessels) were removed to prevent blood contamination. At sacrifice, the mean ± SEM plasma viral load in the SIV-infected macaques was 5.0 × 10^4^ ± 3.2 × 10^4^ RNA copies/mL. SIV-infected macaques (mean ± SEM age: 5.1 ± 0.2 years; mean ± SEM weight: 5.5 ± 1.0 kg) were compared with noninfected controls (mean ± SEM age: 9.5 ± 1.8 years; mean ± SEM weight: 12.1 ± 2.0 kg) housed under similar conditions.

### 2.2. Isolation of ASCs, Cell Culture, and Cell Treatments

The experimental procedures involving human ASCs were approved by the local Ethical Committees for human research and were performed in accordance with the European Union’s guidelines and the Declaration of Helsinki. ASCs were isolated from human abdominal subcutaneous adipose tissue (SCAT) from nine healthy women (mean ± SEM age: 42.9 ± 3.9 years; mean ± SEM BMI: 24.2 ± 0.8 kg/m^2^) by liposuction during plastic surgery. Donors were nonobese and nondiabetic and had no history of HIV, HCV, or HBV infection. Briefly, SCAT samples were digested with collagenase, filtered to remove large debris, and cultured at an initial density of 5 × 10^4^ cells/cm^2^ in alpha-minimum essential medium (αMEM) supplemented with 10% fetal bovine serum (Gibco, Invitrogen Corporation, San Diego, CA, USA), 2 mmol/L glutamine, 2.5 ng/mL basic fibroblast growth factor (PeproTech, Rocky Hill, NJ, USA), and penicillin/streptomycin (Gibco, Invitrogen Corporation, San Diego, CA, USA), as described previously [9]. After 24 h, nonadherent cells were removed and the medium was changed. Cultures were fed every two to three days and trypsinized and passaged every five days. Cells were exposed (or not) to 40 nmol/L recombinant HIV proteins Tat or Nef (Jenabioscience, Jena, Germany) at clinically relevant concentrations (i.e., those observed in treatment-naïve, HIV-infected patients [9,23,24,25]) for up to 30 days, between passages three and nine. In some experiments, ASCs were exposed to HIV proteins to 250 µmol/L *N*-acetyl-cysteine (NAC) (Sigma-Aldrich, St Louis, MO, USA): After 15 days of treatment with HIV proteins, we started a concomitant NAC treatment for 10 days. Thus, the impact of NAC was analyzed at day 25.

### 2.3. Evaluation of Cell Proliferation

ASC proliferation was evaluated in terms of the cell population doubling level (PDL, calculated as described previously [19]), where D0 and D5 are the cell counts at seeding and harvesting, respectively:PDL = log2 (D5/D0)(1)

### 2.4. Adipocyte Differentiation

After 30 days of treatment with Tat or Nef, differentiation of the ASCs into adipocytes was triggered for 14 days in the absence of HIV proteins. ASCs were cultured for six days in adipogenic induction medium (1 µmol/L dexamethasone, 500 µmol/L 3-isobutyl-1-methylxanthine, 1 µmol/L insulin, and 1 µmol/L rosiglitazone) and then cultured in adipogenic maintenance medium (1 µM insulin). On day 14, cells were stained with Oil-Red-O (Sigma-Aldrich), as described previously [26]. The Oil-Red-O staining level was quantified as the absorbance at 520 nm and normalized against the protein content.

### 2.5. Cellular Senescence Assay

The blue staining produced by SA-β-galactosidase’s hydrolysis of 5-bromo-4-chloro-3-indolyl-β-d-galactopyranoside (X-Gal, Sigma-Aldrich, St Louis, MO, USA) was used as a biomarker of cellular senescence. To quantify SA-β-galactosidase activity, cells were incubated with appropriate buffer solution containing X-Gal at pH 6, as described previously [19]. The proportion of positive (blue) cells was determined. We used the acidotropic dye LysoTracker (Invitrogen Corporation) to evaluate lysosomal mass. Briefly, cells were cultured in 96-well plates and incubated with LysoTracker (50 nmol/L) in αMEM for 2 h at 37 °C in the dark. The results were quantified on a plate fluorescence reader (Spectrafluor Plus, Tecan France, Trappes, France) at 504–570 nm, and normalized against total nuclear DNA (stained with 4,6-diamidino-2-phenylindole (DAPI) at 345–458 nm).

### 2.6. Protein Secretion Assay

The levels of human IL-6 and IL-8 secreted by ASCs after 15 days of incubation with Tat or Nef were determined in the culture medium from the last 24 h, using Quantikine sandwich ELISAs (R&D Systems, Inc. Minneapolis, MN, USA). The assay sensitivity was 0.7 pg/mL for IL-6 and 3.5 pg/mL for IL-8. Total levels of human adiponectin/Acrp30 and leptin secreted by differentiated ASCs after 14 days of differentiation were determined in the culture medium from the last 24 h, using Quantikine sandwich ELISAs (R&D Systems). The assay sensitivity was 0.246 ng/mL for total adiponectin/Acrp30 and 7.8 pg/mL for leptin.

### 2.7. Assays of Oxidative Stress and Mitochondrial Dysfunction

The production of reactive oxygen species (ROS) was assayed by quantifying the oxidation of 5-6-chloromethyl-2,7-dichlorodihydro-fluorescein diacetate (CM-H_2_DCFDA, Invitrogen Corporation) on a plate fluorescence reader at 520–595 nm. Results were normalized against DAPI fluorescence, as described previously [19]. ROS production was also assessed by the oxidation of nitro blue tetrazolium (NBT, Sigma-Aldrich), as quantified by the absorbance at 560 nm and normalized against the protein content. To evaluate mitochondrial dysfunction, we used the cationic dye tetrachloro-tetra-ethyl-benzimidazolyl-carbocyanine iodide (JC-1) to evaluate mitochondrial membrane potential, and the MitoTracker Red probe (both from Invitrogen Corporation) to measure mitochondrial mass, as described previously [19]. Briefly, cells were cultured in 96-well plates and incubated with JC-1 (4 µg/mL) or MitoTracker (50 nmol/L) for 2 h at 37 °C in the dark. The results were quantified on a plate fluorescence reader at 520–595 and 485–535 nm for JC-1 aggregates and monomers, respectively, and at 575–620 nm for MitoTracker.

### 2.8. RNA Isolation and Quantitative RT-PCR

Total RNA was isolated from cultured cells using a RNeasy kit (Qiagen, Valencia, CA, USA) and mRNA expression was analyzed using RT-PCR, as described previously [9]. The sequences of the oligonucleotide primers are given in Appendix A.

### 2.9. Protein Extraction and Western Blotting

Proteins were extracted from the VAT and SCAT of macaques and ASC monolayers, as described previously [9], and then electroblotted on a nitrocellulose membrane (Biorad Laboratories, Richmond, CA, USA). Specific proteins were detected by incubation with specific primary antibodies for p16 (BD Pharminogen, Franklin Lakes, NJ, USA, 51-1325GR, dilution 1/500), phosphorylated-p53 (Abcam, Cambridge, UK, ab3897, dilution 1/1200), total p53 (Abcam, ab1101 dilution 1/1200), phospho-Akt (Ser473, Cell Signaling, Danvers, MA, USA, cs9271, dilution 1/1000), Akt (Cell signaling, cs9272, dilution 1/1000) and tubulin (Sigma, T5168, dilution 1/10,000), and then with horseradish-peroxidase-conjugated secondary antibodies at 1/5000 of dilution except for tubulin, the secondary antibody was diluted at 1/10,000. Immune complexes were detected by enhanced chemiluminescence (Amersham, GE Healthcare Europe, Velizy-Villacoublay, France).

### 2.10. Statistical Analysis

Experiments were performed at least three times on triplicate samples. Data are expressed as the mean ± SEM. Statistical significance for HIV-protein-treated cells vs. control cells with or without NAC, and for SIV-infected vs. control macaques were determined with a parametric test (Student’s t-test) or a nonparametric test (the Mann–Whitney U-test), as appropriate. We also performed a two-way ANOVA to study the impact of adipose tissue localization on senescence markers in our in vivo model.

## 3. Results

### 3.1. Adipose Tissue in SIV-Infected Macaques Displays Features of Senescence

We observed that VAT and SCAT p16 expression was higher in infected macaques than in control macaques (Figure 1A). Furthermore, the expression of phosphorylated-p53 (normalized against total p53 and tubulin) was higher in SCAT and similar in VAT of SIV-infected macaques, when compared with uninfected macaques (Figure 1B). We found a significant effect of SIV infection (*p* = 0.0009) and of adipose tissue localization (*p* = 0.05) for p16 expression. Thus, according to our results, the greater expression of p16 in VAT, suggests that VAT displays a higher aging phenotype. Moreover, p16 level and p53 activation in SCAT or VAT did not correlate with viral load, suggesting that the level of senescence was not linked to the severity of SIV infection. These results indicate that adipose tissue was more senescent in infected macaques and strongly suggest that SIV per se accentuates the aging of adipose tissue.

### 3.2. Tat- and Nef-Induced Cell Senescence in ASCs

#### 3.2.1. Treatment of ASCs with Tat and Nef Resulted in a Lower Proliferative Capacity and Higher Levels of Senescence Markers

Next, we looked at whether the HIV proteins Tat and Nef could induce senescence in ASCs. To this end, we first determined the impact of up to 30 days of exposure to Tat or Nef on cell proliferation in vitro. We found that Tat and Nef lowered the ASCs proliferation rate. This effect was seen after 15 days, and the low proliferation rate fell further with each cell passage (Figure 2A), when compared with nontreated cells. After 20 days of treatment, the cumulative PDL was significantly lower in ASCs treated with Tat or Nef than in nontreated cells. On day 15, the two HIV proteins enhanced senescence in ASCs, as characterized by a higher senescent cell count (based on the SA-β-galactosidase activity). The percentage of senescent cells was 15.6 ± 1.3% and 19.3 ± 2.1% for Tat- and Nef-treated cells respectively, vs. 10.4 ± 1.1% for control cells (Figure 2B). Furthermore, treatment with the HIV proteins was associated with greater lysosome accumulation (Figure 2C). Lastly, the expression of the cell cycle arrest proteins p16 and the level of p53 activation were higher after 15 days of Tat and Nef treatment, relative to controls (Figure 2D). Tat- or Nef-treated ASCs displayed signs of SASP, with greater secretion of the pro-inflammatory cytokines IL-6 and IL-8 (Figure 3A,B). Taken as a whole, these data indicated that treatment with the HIV protein Tat or Nef induced the cellular senescence in ASCs. In general, Nef had a greater effect than Tat on the induction of senescence and the secretion of inflammatory proteins.

#### 3.2.2. Tat- or Nef-Induced Oxidative Stress and Mitochondrial Dysfunction in ASCs

ROS production (measured through NBT or CM-H2-DCFDA oxidation) was elevated in Tat- or Nef-treated ASCs (Figure 4A). Using MitoTracker, we observed that the mitochondrial mass was 1.3- and 1.5-fold higher than in control cells after 15 days of treatment with the HIV proteins Tat or Nef, respectively (Figure 4B left panel). We also observed a concomitant destabilization of the mitochondrial membrane potential, as shown by the relative decrease in JC1 aggregation (Figure 4B right panel). Despite the higher mitochondrial mass, Tat- or Nef-treated cells had a mitochondrial membrane potential that was 30% lower than in controls; this was suggestive of mitochondrial dysfunction. Overall, these results indicate that the cellular senescence induced by 15 days of treatment with the HIV proteins Tat or Nef was associated with higher ROS production, which might have been due to mitochondrial dysfunction. Again, Nef tended to have a greater effect than Tat.

#### 3.2.3. Inhibition of Oxidative Stress Prevented Tat- and Nef-Induced Senescence in ASCs

To assess the involvement of oxidative stress in Tat- and Nef-induced senescence, ASCs were treated with Tat or Nef for 15 days and then the antioxidant NAC was added (or not) for further 10 days. Experiments were performed on day 25. NAC suppressed the elevation in ROS production induced by Tat or Nef, as shown by NBT and CM-H_2_-DCFDA oxidation (Figure 5A). NAC also prevented the relative decrease in cell proliferation induced by Tat or Nef (Figure 5B). Accordingly, NAC decreased the percentage of senescent cells (Figure 5C) and the mitochondrial mass in Tat- or Nef-treated cells (Figure 5D). The same trend was observed for the mitochondrial membrane potential. These results indicate that oxidative stress was probably involved in the senescence induced by Tat and Nef.

### 3.3. Adipogenesis is Impaired in ASCs Displaying Nef-Induced Senescence

To assess the adipogenic potential of senescence ASCs, cells were induced to differentiate into adipocytes after 30 days of treatment with Tat or Nef. After 14 days of differentiation, treatment with Nef (but not Tat) resulted in a lower level of lipid accumulation (Figure 6A) and lower expression levels for the adipogenic marker genes *PPARG, CEBPA,* and *FABP4* (Figure 6B). Furthermore, we observed that Nef-induced senescence was associated with lower secreted levels of the adipokines leptin and adiponectin in the culture medium (Figure 6C).

Taken as a whole, adipogenic fate was affected in ASCs presenting Nef-induced senescence. The absence of an effect on Tat-treated ASCs in this setting might have been due to the protein’s weaker ability to induce senescence, inflammation, and oxidative stress.

### 3.4. Treatment of ASCs with Nef or Tat is Associated with Insulin Resistance in ASC-Derived Adipocytes

In order to assess the functional status of adipocytes derived from Tat- or Nef-treated ASCs, we evaluated the cells’ insulin sensitivity. A Western blot analysis showed that Tat- and Nef-treated cells expressed similar, normal levels of total Akt the key enzyme in the insulin signaling pathway, and mainly involved in short-term metabolic responses (Figure 7A). Despite these normal total Akt levels, the activation of Akt in response to acute stimulation by insulin was 24% and 22% below control values in adipocytes derived from Nef- or Tat-treated ASCs (Figure 7B), indicating a lower level of insulin sensitivity.

## 4. Discussion

Our present results showed that SIV/HIV infection was associated with certain features of adipose tissue aging. Firstly, fat from SIV-infected macaques displayed markers of senescence. Since adipose tissue has been identified as an HIV reservoir that contains infected cells (even in ART-controlled patients [20,21,27]), we hypothesized that HIV proteins released by infected cells might exert a bystander effect on neighboring, noninfected cells (mainly adipocytes and their precursors, such as ASCs). Our study is the first to have shown that the HIV proteins Tat and Nef can enhance senescence in ASCs, and that this effect could be partially reversed by treatment with an antioxidant. Furthermore, we found that Nef-induced senescence was associated with an abnormally low adipogenic capacity, and impaired adipocyte function.

To the best of our knowledge, the level of senescence in the adipose tissue of SIV-infected macaques or in ART-naïve, HIV-infected people has not previously been evaluated. Here, we showed that SIV infection is associated with higher adipose tissue levels of p16 in VAT and SCAT and activated p53 in SCAT (both hallmarks of aging [22]), which suggests that senescent cells are present in adipose tissue. The discrepancy between p16 expression and p53 activation in VAT may be due to the fact that senescence can involve either the p53–p21 or the canonical p16–retinoblastoma protein tumor suppressor pathways [28]. We propose that in VAT the p16 pathway of senescence [29] is activated in priority by SIV and its proteins rather than that using p53. Accordingly, recent data show that aging is not associated with p53 activation in VAT [30]. We found a significant effect both of SIV infection and of adipose tissue localization on p16 expression. Thus, according to our results, the greater expression of p16 in VAT, suggests that VAT displays a higher aging phenotype than SCAT.

Adipose tissue is an HIV/SIV reservoir even in the presence of suppressive ART [20,21,27]. Furthermore, the level of HIV proteins, such as Tat or Nef, is measurable in the plasma and in tissues of HIV-infected ART-treated patients [24,31,32]. Thus, Tat was detected in intestinal epithelium from ART-treated patients [31]. As well, Nef was detected in cardiomyocytes of SIV-infected ART-treated macaques [32]. These studies support the hypothesis that HIV proteins could be present in AT and could have a deleterious effect on nearby noninfected ASC and adipocytes by a “bystander effect”. Previous studies have reported that Tat and Nef can induce cellular senescence and dysfunction in endothelial cells [33] and in bone marrow MSCs [19]. Accordingly, we found that in vitro incubation with the HIV proteins Tat and Nef lowered ASCs’ proliferative capacity and resulted in higher levels of several senescence biomarkers, including SA-beta-galactosidase activity, cell cycle arrest proteins (p16 and phosphorylated p53), and SASP proteins (secreted IL-6 and IL-8). These are the first results showing a link between the virus itself and aging in adipose tissue. There was no evidence for a link between senescence and the severity of SIV infection. Otherwise, we cannot assess the presence of a direct link between Tat or Nef and higher senescence in our in vivo model. Nevertheless, our in vitro results indicated that Tat and Nef induced senescence in ASCs that could in turn participate to the aging of adipose tissue.

With regard to the mechanisms that might promote senescence in ASCs, greater mitochondrial dysfunction and ROS production have previously been linked to the occurrence of age-dependent senescence in ASCs from individuals not infected with HIV [34]. However, it has been shown that adipose tissue from ART-naïve, HIV-infected patients displayed signs of mitochondrial toxicity and higher levels of oxidative stress—indicating a probable ART-independent mechanism of action for HIV and/or HIV proteins in the occurrence of mitochondrial dysfunction [35,36]. It has also been reported that Tat induces oxidative stress in T-cells and other cell types [37,38], and that Tat and Nef alter mitochondrial function in neurons [38]. Accordingly, we showed that the onset of senescence in ASCs correlated with low mitochondrial activity and high ROS production. Importantly, treatment with the antioxidant NAC partially countered the negative impact of HIV proteins; this finding highlighted the role of oxidative stress in Tat- and Nef-induced senescence. The ROS induced by Tat or Nef treatment of ASCs might be produced in the mitochondria or by NADPH oxidase activity [14]. NAC is a powerful antioxidant that directly scavenges oxygen radicals from all sources. Given that Tat and Nef treatments were associated with a low mitochondrial membrane potential in our experiments, the ROS were probably generated in dysfunctional mitochondria [14]. To the best of our knowledge, the present study is the first to have shown that the HIV proteins Tat and Nef induced senescence in ASCs and that the latter might result (at least in part) from the induction of ROS production.

We observed that Tat and Nef induced a pro-inflammatory profile in ASCs. The profile might contribute to inflammaging, i.e., a low-grade, chronic, systemic inflammatory state that can be induced by oxidative stress or by pathogens. In turn, inflammaging might enhance the aging of adipose tissue. Moreover, inflammation can trigger senescence and promote fibrosis (collagen production) [39,40]. We have previously reported that (i) HIV/SIV infection induced collagen deposition in adipose tissue and (ii) HIV proteins induced a pro-fibrotic phenotype in ASCs in vitro [9]; this fits with the ASCs pro-inflammatory profile observed in the present study. Oxidative stress was shown to induce adipose tissue senescence and fibrosis in a mouse model [41]. We hypothesize that HIV proteins induced oxidative stress, which in turn led to cellular senescence and fibrosis. However, the putative causal link between fibrosis and senescence requires further investigation.

One of the main characteristics of aging is a decline in the tissues’ regenerative potential. Adipose tissue is a source of multipotent stem cells, namely ASCs [1,42]. Therefore, a decline in ASC proliferation and/or function (reflected by cell senescence) might contribute to a loss of adipose tissue homeostasis—especially through increased SASP factor secretion. Senescence induced by Tat and Nef might contribute to ASC exhaustion in the adipose tissue of HIV-infected people.

It is known that senescence of adipose tissue precursor cells is associated with low adipogenic marker expression and alterations in adipogenesis [1,11,43,44]. Nef can interact with the adipocyte master regulatory protein PPARγ and suppress its activity, leading to low expression of genes involved in lipid accumulation [45,46]. Tat’s impact is less clear; depending on the cellular model studied, it can either inhibit or promote adipogenesis [9,47,48]. In the present study, only Nef-induced senescence was associated with impairments in adipogenesis: lipid accumulation, the mRNA expression of pro-adipogenic markers (such as *PPARG*, *CEBPA*, and *FABP4*), adipokine secretion, and insulin sensitivity. By removing Tat or Nef during ASC differentiation, we ruled out a direct effect of these HIV proteins on adipogenesis. Thus, our data suggest that the senescence and dysfunction induced by HIV proteins during ASC proliferation partially suppressed adipogenesis. The present results are in line with our previous work, in which we showed that HIV/SIV infection was associated with the presence of smaller adipocytes in both SIV-infected macaques and HIV-infected people [9]. Our results are also in line with the observation that the adipose tissue of ART-naïve HIV-infected individuals contains low levels of PPARγ and adipose-specific markers and high levels of markers of mitochondrial dysfunction [36].

Low adipogenic potential in ASCs might be involved in metabolic impairments, such as insulin resistance [1,42]. Given that ASC acquire insulin sensitivity during differentiation, a decline in their adipogenic potential might drive insulin resistance by limiting adipose tissue expansion [44]. We showed here that Nef- and Tat-induced senescence is associated with low insulin sensitivity. Thus, in the HIV-infected population, aging of the adipose tissue might be involved in the onset of insulin resistance and greater susceptibility to diabetes [4,49,50,51,52]. Several studies have shown that HIV infection is associated with atherosclerosis [53]; the fact that this is also observed in the absence of treatment (i.e., HIV controllers) [54] emphasizes virus’s role in the onset of cardiometabolic complications. Further studies are required to define the role of HIV-induced adipose tissue senescence in the impairment of fat distribution and function and in the cardiometabolic complications observed in aging HIV-infected individuals.

The main objective of our study was to evaluate, at first, the impact of SIV/HIV infection per se, on adipose tissue in the absence of ART. In addition to HIV infection, the role of different ART molecules on adipose tissue/adipocytes has been widely evaluated both in vivo and in vitro. First generation thymidine nucleotide reverse transcriptase inhibitors (NRTIs: Stavudine, zidovudine) were associated with the development of clinical lipoatrophy. They were also shown to induce cellular senescence, mitochondrial toxicity and oxidative stress in vitro [26], and mitochondrial dysfunction in the adipose tissue from HIV-infected lipodystrophic patients [55]. First generation PIs induced cellular senescence in fibroblasts [17], endothelial cells [56,57,58], and in smooth muscle cells [15]. Furthermore, PI-treated patients presented accumulation of the senescent marker prelamin A and increased fibrosis in cervical SCAT suggesting a link between fibrosis and aging [8]. Several studies reported a higher level of fibrosis in adipose tissue of ART-treated HIV-infected patients compared to noninfected subjects [6,7,9]. Finally, integrase inhibitors (INSTIs), a recent widely used class of ART, have less impact on senescence of endothelial cells together with anti-inflammatory properties [58,59] but their use is associated with fat gain and higher AT fibrosis [60,61,62,63]). Taken as a whole, ART can also have a role in adipose tissue aging in HIV-infected patients.

The present study had several limitations. As described above, we found high levels of senescence markers in whole adipose tissue from SIV-infected macaques. Although we showed that ASCs entered senescence in response to HIV proteins in vitro, it is still possible that the senescence observed in adipose tissue in vivo is related to the expression of markers by other adipose tissue cells. Our macaque study has some limitations including the range of duration of chronic infection and the older age of the control group. However, we observed higher features of aging in the adipose tissue of SIV-infected macaques which were younger. We could only show association between Tat and Nef proteins and senescence in vitro. In our in vitro study, we used HIV protein concentration measured in the plasma of naïve HIV-infected subjects which could be higher that the concentration in ART-treated patients. Nevertheless, these are the first results showing that there is a link between the virus itself and aging adipose tissue. Given that our in vitro results indicated that Tat and Nef induced senescence in ASCs, we suggest that there could be a causal link between the virus, HIV proteins, and aging of adipose tissue.

Although the senescence in ASCs induced by Nef was associated with impaired adipogenesis, this was not the case for Tat. We hypothesize that Nef’s greater effect on the level of senescence, oxidative stress, and inflammation suggests that there is a threshold for the harmful impact of these changes; however, we did not prove this hypothesis. Furthermore, we did not analyze the impact of NAC on the ASCs’ ability to differentiate into adipocytes. Lastly, it would be interesting to further evaluate the probable relationships between senescence, inflammation, and fibrosis. Since we wanted to study the effect of SIV/HIV on fat in the absence of treatment, we evaluated SIV infection in macaques and the effect of HIV proteins in vitro; however, we did not study adipose tissue obtained from ART-naïve HIV infected patients.

## 5. Conclusions

SIV/HIV infection induces senescence and associated disorders (including oxidative stress, inflammation, and metabolic alterations) in adipose tissue. These changes might be involved in the alterations in fat redistribution and the cardiometabolic diseases observed in aging HIV-infected individuals.

## Figures and Tables

**Figure 1 cells-09-00854-f001:**
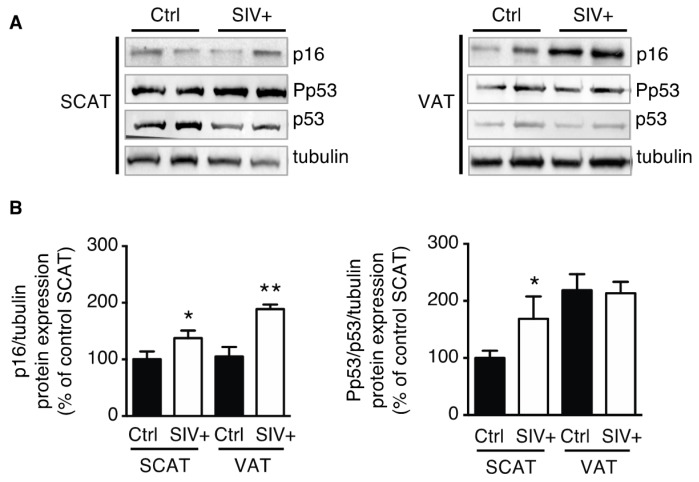
SIV infection of macaques was associated with higher expression of p16 and greater activation of p53 in the adipose tissue. Whole-tissue proteins were extracted from subcutaneous adipose tissue (SCAT) and visceral adipose tissue (VAT) from chronically infected macaques and controls and then analyzed by immunoblotting. (**A**) Representative immunoblots of p16, phosphorylated-p53, p53, and tubulin (loading control) are shown. (**B**) Densitometry analyses against tubulin as loading control were performed for p16 and p53 activation, and expressed as a mean ± SEM. Experiments were performed using SCAT and VAT from macaques from three control uninfected macaques (Ctrl) and four SIV-infected macaques (SIV+). * *P* < 0.05, ** *P* < 0.01 vs. noninfected macaques.

**Figure 2 cells-09-00854-f002:**
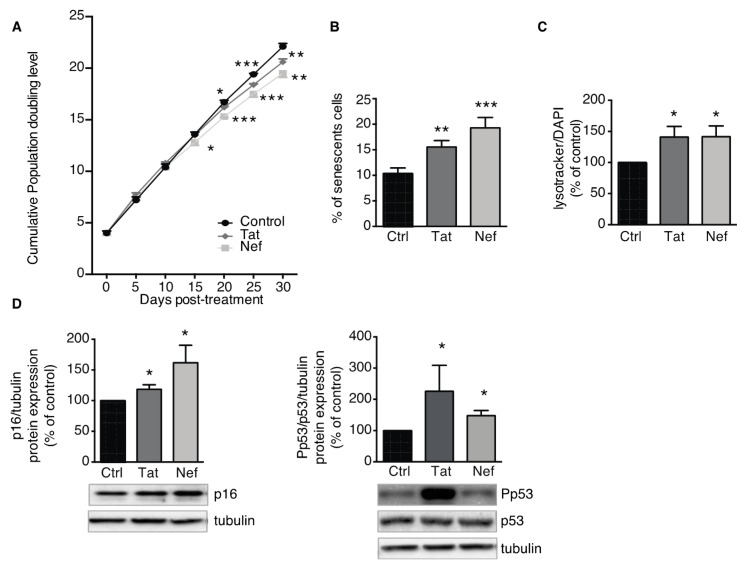
Trans-activator of transcription (Tat) and negative-regulating factor (Nef) proteins induce cell senescence in adipose stem cells (ASCs). ASCs, isolated from different abdominal SCAT healthy donors, were cultured with the HIV proteins Tat or Nef for 30 days. (**A**) Calculation of the cumulative population doubling level (PDL) is described in the Material and Methods Section. Mean ± SEM PDL values were determined on the indicated days of treatment with the HIV proteins (*n* = 9, in duplicate). (**B**) After 15 days of treatment with Tat or Nef, senescence was evaluated in terms of SA-β-galactosidase activity at pH 6 and expressed as the number of SA-β-galactosidase-positive cells as a percentage of total cells (*n* = 9). (**C**) Lysosomal accumulation was assessed as LysoTracker fluorescence, normalized against DAPI (*n* = 9, in duplicate). (**D**) Whole-cell lysates were extracted from ASCs after 15 days of treatment with Tat or Nef, and analyzed by immunoblotting. Representative immunoblots of cell cycle arrest markers p16, phosphorylated-p53, and p53 and tubulin (loading control) are shown (*n* = 4). Densitometry analyses against tubulin as loading control were performed for p16 and p53 activation, and expressed as a mean % of the value for control cells ± SEM. * *P* < 0.05, ** *P* < 0.01, *** *P* <0.001 vs. control cells.

**Figure 3 cells-09-00854-f003:**
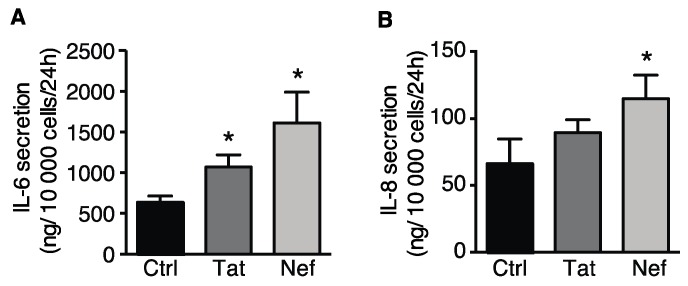
Elevated levels of interleukin (IL)-6 and IL-8 secreted into the culture medium of Tat- or Nef-treated ASCs. After 15 days of treatment with Tat or Nef, the levels of (**A**) IL-6 and (**B**) IL-8 in the culture medium from the last 24 h were determined with ELISAs (*n* = 3, in duplicate). The results are expressed as the mean ± SEM. Experiments were performed in duplicate on ASCs isolated from different abdominal SCAT donors. * *P* < 0.05 vs. control cells.

**Figure 4 cells-09-00854-f004:**
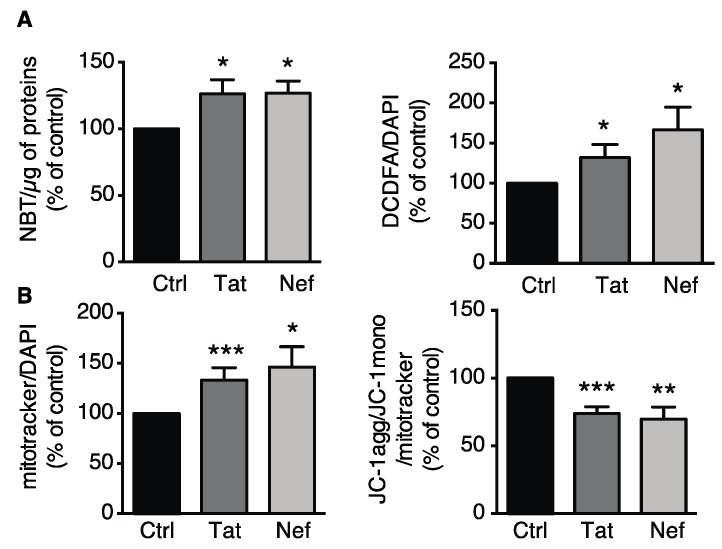
Treatment of ASCs with Tat or Nef resulted in higher oxidative stress and mitochondrial dysfunction. (**A**) After 15 days of treatment with Tat or Nef in ASCs, isolated from different abdominal SCAT healthy donors, reactive oxygen species (ROS) production was assessed spectroscopically by measuring the nitro blue tetrazolium (NBT) absorbance (normalized against protein content) and the CM-H_2_DCFDA fluorescence (normalized against DAPI) (*n* = 9, in duplicate). (**B**) Mitochondrial mass was evaluated using MitoTracker Red dye. The cationic dye JC-1 was used to evaluate the mitochondrial membrane potential. The fluorescence results are expressed as the mean ± SEM % JC-1 aggregate/monomer ratio, relative to control cells (*n* = 9, in duplicate). * *P* < 0.05, ** *P* < 0.01, *** *P* < 0.001 vs. control cells.

**Figure 5 cells-09-00854-f005:**
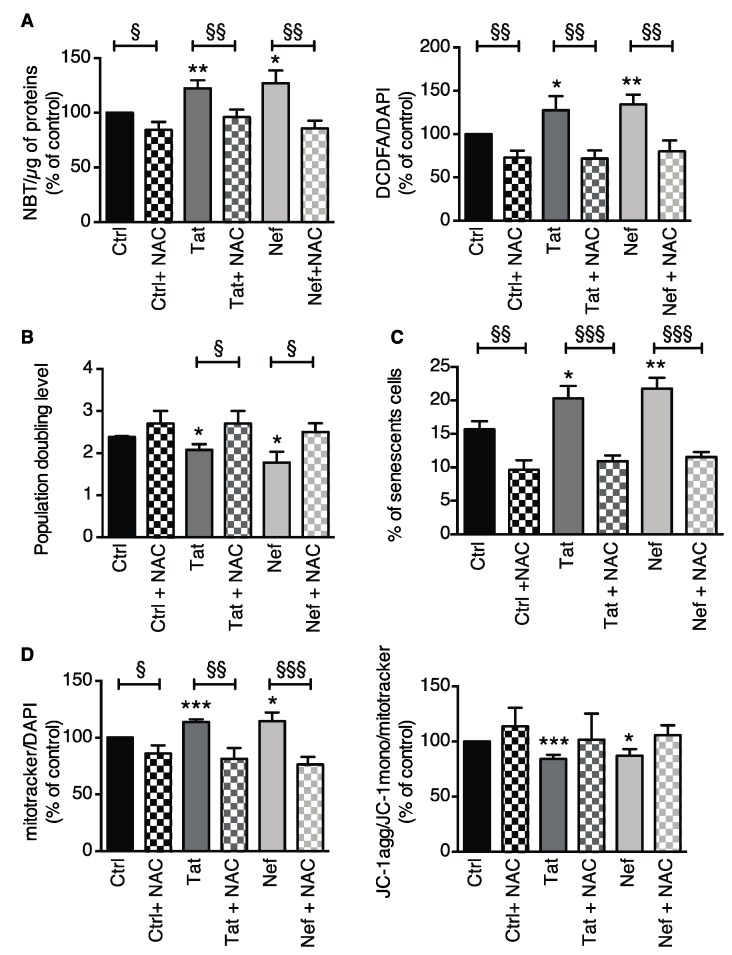
The suppression of oxidative stress using NAC prevented Tat- or Nef-induced senescence in ASCs. ASCs were treated with the HIV proteins Tat or Nef for 15 days. After these 15 days of treatment, we started a concomitant NAC treatment for 10 days. Experiments were performed on day 25 in ASCs, isolated from different abdominal SCAT healthy donors. (**A**) ROS production was assessed spectroscopically by measuring the NBT absorbance (normalized against protein content) and the CM-H_2_DCFDA fluorescence (normalized against DAPI) (*n* = 5, in duplicate). (**B**) The population doubling level (PDL) was calculated as described previously (*n* = 5, in duplicate). (**C**) Senescence was evaluated in terms of SA-β-galactosidase activity at pH 6 and expressed as the proportion (in %) of SA-β-galactosidase-positive cells (*n* = 5, in duplicate). (**D**) Mitochondrial mass was evaluated using MitoTracker dye. The cationic dye JC-1 was used to evaluate the mitochondrial membrane potential. The fluorescence results are expressed as the mean ± SEM % JC-1 aggregate/monomer ratio. The results were normalized against DAPI fluorescence, and expressed as the mean ± SEM % of control cells (*n* = 5, in duplicate). * *P* < 0.05, ** *P* < 0.01, and *** *P* < 0.001 vs. control cells. ^§^
*P* < 0.05, ^§§^
*P* < 0.01, and ^§§§^
*P* < 0.001 NAC-treated vs. nontreated cells.

**Figure 6 cells-09-00854-f006:**
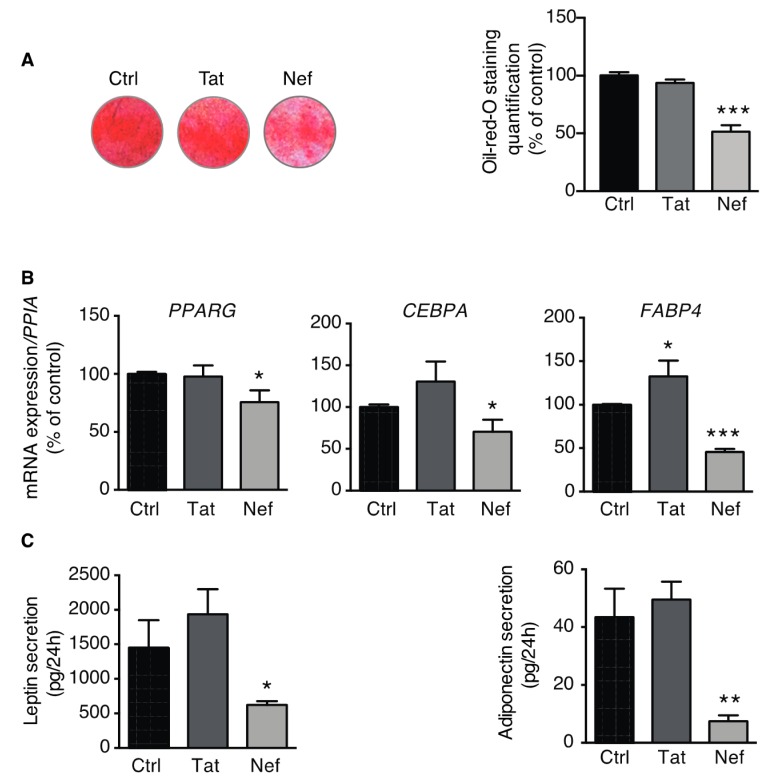
Nef-induced senescence was associated with impaired adipogenesis. After 30 days of treatment with Tat or Nef, ASCs, isolated from different abdominal SCAT healthy donors, were induced to differentiate into adipocytes for 14 days. The Tat or Nef treatment was stopped during differentiation. (**A**) Adipogenic potential of ASCs was determined using Oil-Red-O staining 14 days after the start of differentiation induction. Representative pictures of cultured cells are shown (left panel), and the intensity of Oil-Red-O staining was normalized against protein content and expressed as the mean ± SEM % of control cells (*n* = 4, in duplicate). (**B**) The relative mRNA expression levels of *PPARG, CEBPA,* and *FABP4* were normalized against that of *PPIA*. The results are expressed as the mean ± SEM % of control cells (*n* = 4, in duplicate). (**C**) Secreted leptin and adiponectin levels in the culture medium from the last 24 h of incubation were determined using ELISAs (*n* = 3, in duplicate). The results are expressed as mean ± SEM. * *P* < 0.05, ** *P* <0.01, **** *P* < 0.0001 vs. control cells.

**Figure 7 cells-09-00854-f007:**
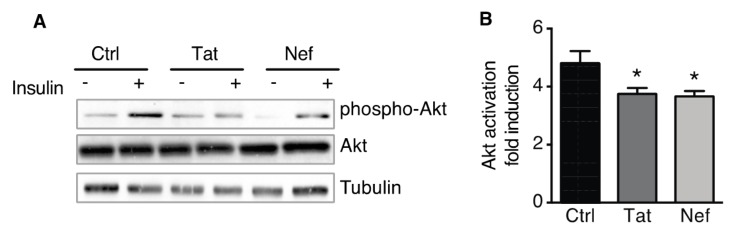
Nef- and Tat-induced senescence in ASCs was associated with insulin resistance in ASC-derived adipocytes. After 30 days of treatment with Tat or Nef, ASCs, isolated from different abdominal SCAT healthy donors, were differentiated into adipocytes for 14 days. The treatment was stopped during the differentiation. (**A**) On day 14 of differentiation, the cells were stimulated (or not) with 100 nmol/L insulin for 7 min. Representative immunoblots of phosphorylated-Akt, Akt, and tubulin (loading control) are shown (*n* = 3). (**B**) The fold induction of Akt activation by insulin was quantified, and results are shown as the mean ± SEM (*n* = 3). * *P* < 0.05, ** *P* < 0.01 vs. control cells.

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
