# Peer review of "SIV Infection and the HIV Proteins Tat and Nef Induce Senescence in Adipose Tissue and Human Adipose Stem Cells, Resulting in Adipocyte Dysfunction"

_cells, 2020, doi:10.3390/cells9040854_

Round 1

Reviewer 1 Report

This manuscript presented results on the adipose dysfunction during SIV infection of macaques in vivo and  the impacts and potential mechanism if Tat and Nef on the  senescence in Adipose Tissue and human adipose stem cells. While this study is clearly interesting, there are some major issues. 

1)  Since the duration and viral loads of SIV infected macaques had wide SD, detailed information of VL, CD4 count for individual macaque should be provided, Further, their correlations with measured AT senescence parameters should be analysed. 

2) The  authors introduced enhanced HIV commodities in infected individuals on oppressive  ART in the paper introduction, authors did not provide data or even discussion about the relevance of Tat and Nef in this context, where the virus is fully suppressed and Tat and Nef expression will be either absent or limited. 

3) SIV infected macaque at chronic stage with viremia have many pathological changes due to SIV infection, how to link the increased senesence of AT to Tat and Nef. 

4) the human donor information, especially basic health condition should be provided, as the   adipose stem cells were derived from these donors, Any basic conditions could impact the results. Download Manuscript

Reviewer 2 Report

This is a well designed study, and I only have minor concerns.

Minor concerns:

Provide antibody dilutions for Western blotting

A limitation of the study is the lack of ART administration to the macaques.  Please include ART effects on senescence in the discussion section

An additional mention of euthanasia criteria for the SIV-infected macaques is warranted.  Why 482+/-118 days?  How long did the controls remain on study. Also, the age difference between the infected and uninfected animals is quite large.

Lipoaspiration was used to collect human AT. Please state whether this is SQ or VAT in the methods.

Since multiple adipose depots were measured, statistical analysis should include 2-way ANOVA results. This same critique holds true for the +/-NAC experiments. The current statistics are correct for the results reported, but it would be interesting to see if there is a depot effect in vivo

The NAC effect looks to be the same regardless of treatment. At what point prior to oxidative stress measures was the NAC withdrawn? This should be added in the methods.

Reviewer 3 Report

The manuscript “SIV Infection and HIV Proteins Tat and Nef Induce Senescence in Adipose Tissue and Human Adipose Stem Cells Resulting in Adipocyte Dysfunction” by Jennifer Gorwood, et al., is of interest to people working in this field. While the study has merit, there were numerous methodological problems that diminish the significance of the results.

Here are a few suggestions to improve the manuscript:

1.The authors indicate in the Material and Methods section that experiments were performed on 7 macaques that were infected or not with SIVmac251. It is not clear how many macaques were infected and how many were not infected. The authors should specify the total number of macaques used in each group.

2.Figure 1: The authors indicate that SCATs and VATs used for this experiment were derived from 3 to 4 different macaques per group. Were the SCATs and VATs isolated from different animals and pooled together? If so, what is the rationale for pooling them together. Data derived from individual macaques together with their corresponding viral loads would be more informative.

3.Data provided in Figure 1, indicating that SIV infection was associated with higher expression of P16 and activation of P53 in the adipose tissue of macaques is not convincing. The data provided in Figure 1A, illustrating the relative protein expression of p16, indicates that the upregulation of p16 expression in VATs isolated from SIV infected macaques has a greater magnitude than in SCATs isolated from SIV infected macaques. However, in Figure 1B, upregulation of hyper-phosphorylated p53 is only observed in SCATs derived from infected macaques. It is not clear why the p16 and pp53 expression in VATS and SCATs isolated from SIV infected macaques are not the same. The authors have not provided any clarification regarding this discrepancy.

4.It is difficult to interpret the quantitation data for the hypo-phosphorylated and hyper-phosphorylated forms of p53 protein provided in Figure 1B and Figure 2D without evaluating loading control data. The loading control data should be included.

5.In Figure 6, the authors indicate that experiments were performed in duplicate or triplicate in ASCs isolated from 3 to 4 different donors. The authors should be precise and clearly state the number of donors used to isolate ASCs. The data provided in Figure 6B, indicates that the downmodulation of PPARG expression in ASCs pretreated with Nef is less than 20% when compared to control ASCs. Why do they consider this to be biologically significant?

6.Data provided in Figure 7A and 7B is not convincing. It is difficult to interpret the data without the loading control.

7.Extensive revisions for grammar, spelling and English language usage should be made.

Round 2

Reviewer 3 Report

The authors have satisfactorily addressed the comments / issues that were raised. The additional data, clarifications and revisions made by the authors significantly improve the quality of the manuscript. This manuscript in the current form is acceptable for publication in Cells.